# Understanding mental health trends during COVID-19 pandemic in the United States using network analysis

**Hiroko Kobayashi**[3,4], **Raul Saenz-Escarcega**[1], **Alexander Fulk**[2], **Folashade B. Agusto**[2]*

**1** Department of Biology, University of Kansas, Lawrence, KS, United States of America, **2** Department of Ecology and Evolutionary Biology, University of Kansas, Lawrence, KS, United States of America, **3** Department of Molecular Biosciences, University of Kansas, Lawrence, KS, United States of America, **4** Department of Psychology, University of Kansas, Lawrence, KS, United States of America

* fbagusto@gmail.com

**Data Availability Statement:** Data used are available at the link https://doi.org/10.1073/pnas. 2111454118.

## Abstract

The emergence of COVID-19 in the United States resulted in a series of federal and state-level lock-downs and COVID-19 related health mandates to manage the spread of the virus. These policies may negatively impact the mental health state of the population. This study focused on the trends in mental health indicators following the COVID-19 pandemic amongst four United States geographical regions, and political party preferences. Indicators of interest included feeling anxious, feeling depressed, and worried about finances. Survey data from the Delphi Group at Carnegie Mellon University were analyzed using clustering algorithms and dynamic connectome obtained from sliding window analysis. Connectome refers to the description of connectivity on a network. United States maps were generated to observe spatial trends and identify communities with similar mental health and COVID-19 trends. Between March 3rd, 2021, and January 10th, 2022, states in the southern geographic region showed similar trends for reported values of feeling anxious and worried about finances. There were no identifiable communities resembling geographical regions or political party preference for the feeling depressed indicator. We observed a high degree of correlation among southern states as well as within Republican states, where the highest correlation values from the dynamic connectome for feeling anxious and feeling depressed variables seemingly overlapped with an increase in COVID-19 related cases, deaths, hospitalizations, and rapid spread of the COVID-19 Delta variant.

## Introduction

Following an initial outbreak in 2019 in Wuhan, China, the novel coronavirus disease (COVID-19) spread rapidly across the world, resulting in nearly 4.2 million infected and approximately 85 thousand dead within the first year of being identified in 2020 [1]. By March 22 of 2020, the World Health Organization (WHO) declared COVID-19 a global pandemic [2]. This declaration resulted in several countries adopting various protocols to collect data from local counties and municipalities to make informed decisions on COVID-19 policies to

**Funding:** This research was funded by the National Science Foundation, grant number DMS2028297 and DMS2230117. The funders had no role in study design, data collection and analysis, decision to publish, or preparation of the manuscript.

**Competing interests:** The authors declare that there are no conflicts of interest.

curb the spread of COVID-19 in their regions [3, 4]. However, due to limited information available on the novel disease, many countries opted to implement large-scale control measures such as lockdowns that had far-reaching impacts on the general population [3].

To manage the spread of COVID-19 in the United States, the federal government and state governments implemented a variety of health mandates [5]. However, the federal government at this time opted for a hands-off approach to COVID-19 and allowed individual states to decide on the best ways to limit the spread of this deadly disease [6]. This variation in leadership led to differing outcomes in terms of the spread of COVID-19 [7]. These mandates to protect the public ranged from school and work closures to stay-at-home orders, all of which can have important effects on several aspects of an individual's life. In addition, the increasing incidence of COVID-19 and COVID-19 related deaths likely increased individual's pandemic related worry [8]. These potential stressors have, and continue to impact each member of the population to varying degrees. Policies were enacted to enforce many decisions related to the pandemic and may have adverse mental health effects including feelings of anxiety and depression. Elevated adverse mental health conditions were reported at a higher disproportionate rate in the second quarter of 2020 (25.5%) compare to the last quarter of 2019 (8.1%) [9, 10]. Several countries, including the United States, have input measures to allow for reporting of psychological distress among the population via surveys to better tailor support and resources for their respective populations [11, 12].

Several studies have concluded that lockdowns and other policies related to COVID-19 can increase mental health burden, especially for vulnerable groups [13–15]. Certain policies related to COVID-19, such as government funds being issued to the general population via stimulus checks, also have the potential for positive impacts on both physical and mental health [16]. Vaccines also likely decreased the prevalence of mental health issues since their initial roll-out in late 2020 [17]. Though a potentially large portion of the public has displayed hesitance towards vaccination and thus may be experiencing similar levels of mental distress [18]. Due to the various possible mental health outcomes that arise from each of these scenarios, it is important to evaluate data in a manner that facilitates dynamic interpretation across time in a clear and concise way.

For instance, Amico and Bulai [19] used network analysis to explore the interactions of COVID-19 among the various regions of Italy and the impact of Italy's governmental policies in response to the spread of SARS-CoV-2. To analyze the network interaction between regions in Italy, they used six indicators (namely the number of hospitalized individuals in IC, number of hospitalized individuals with symptoms, number of individuals in home isolation, number of new positives, number of discharged healed, and number of deceased individuals) to form a correlation network referred to as "Covidome". Their results showed that there was a distinct north-south clustering of the regions in the country. Furthermore, they found distinct differences in the Covidome fluctuations between the first and second waves of the pandemic based on region-specific political choices.

In this study, we apply network and clustering analysis to understand the connectivity between states and how COVID-19 impacts mental health across the United States using COVID-19 related mental health indicators such as feeling anxious, feeling depressed, and worried about finances. We follow the approach in Amico and Bulai [19] and study the covariance matrix of the mental health indicators. Connectome is a term commonly used in the field of neuroscience to examine brain network dynamics [19, 20]; it refers to the complete description of the structural connectivity of an organism's nervous system [21]. In this paper, we use this term to describe the connectivity amongst the states in terms of the three mental health indicators and we refer to it as the "mental health connectome". We compare the results from this study with results from Fulk et al [22] that investigated the effect of several COVID-19

related indicators variables on anxiety and depression in the general population using the same data during the same time period as this study. The rest of the paper is organized as follows: the Methods section gives an overview of the methods used in the analyses, including a description of the data; the Results section describes the results of the study. In the Discussion section we put our results into the context of policies that were implemented and also evaluate how our results compare to Fulk et al [22] analysis of this data, as well as their conclusion.

## Methods

### Mental health related data and policy timeline

**Mental health data.** This research is based on survey results from Carnegie Mellon University's Delphi Group. The Carnegie Mellon University U.S. COVID-19 Trends and Impact Survey (Delphi US CTIS [23]) was distributed in partnership with Facebook in the form of a daily voluntary survey that Facebook users could respond to. Survey questions ranged from those concerning physical health, and the economic impact of COVID-19, to mental health and behavioral prompts. Participant responses were collected, aggregated and de-identified, and made publicly available through Delphi US CTIS open access dataset. The Delphi US CTIS data was accessed through the Covidcast library.

Since the data used was aggregated and de-identified, we did not seek approval from the University of Kansas ethics committee. In this study, three indicators of interest were selected to represent the impact of COVID-19 on mental health. These three indicators were (i) the percentage of participants who experienced feelings of anxiety within the last 7 days, (ii) the percentage of participants who felt worried about their finances for the following month, and (iii) the percentage of participants who felt depressed within the last 7 days. We refer to these three indicators as feeling anxious, feeling depressed, and worried about finances throughout this paper. Daily confirmed COVID-19 cases and daily deaths from the Center for Systems Science and Engineering at Johns Hopkins University [24] were also included in this study. These indicators were used to assess whether there was any relationship between trends in cases or deaths and the trends in mental health indicators.

The survey results collected can be categorized into two separate time frames, the first starting from September 8th, 2020 running until March 2nd, 2021, and the second beginning from March 2nd, 2021 to January 10th, 2022. This separation is due to a change in the survey questions asked about anxiety and depression; in the first time frame, the survey asked participants whether they experienced feelings of anxiety or depression for the past 5 days. From March 2nd, 2021, the survey asked whether the participants experienced feelings of anxiety or depression for the past 7 days, resulting in data collected from March 2nd differing from that of earlier dates. Data analysis was only conducted using data collected from the second time frame due to missing data points for mental health indicators from the first time frame.

**Data presentation.** To see any mental health trends and similarities among states we grouped the data in two different ways, geographically and politically. Fig 1(a) shows the geographic map of the country divided into four regions: namely midwest, northeast, south, and west. There are different ways in which the country can be split geographically; in many cases, it might be based on the historical relationship within certain regions. These relationships can stem from similar religious communities, cultural similarities, shared historical significance, or similar climates in the region [25]. In this study, we use geographical grouping from the United States Census Bureau [26]. Fig 1(b) depicts the map of the country along political party preferences, namely Democratic and Republican party preferences. The political party preference was determined from the results of the 2020 presidential election using the senate election, and overall state party registration [27].

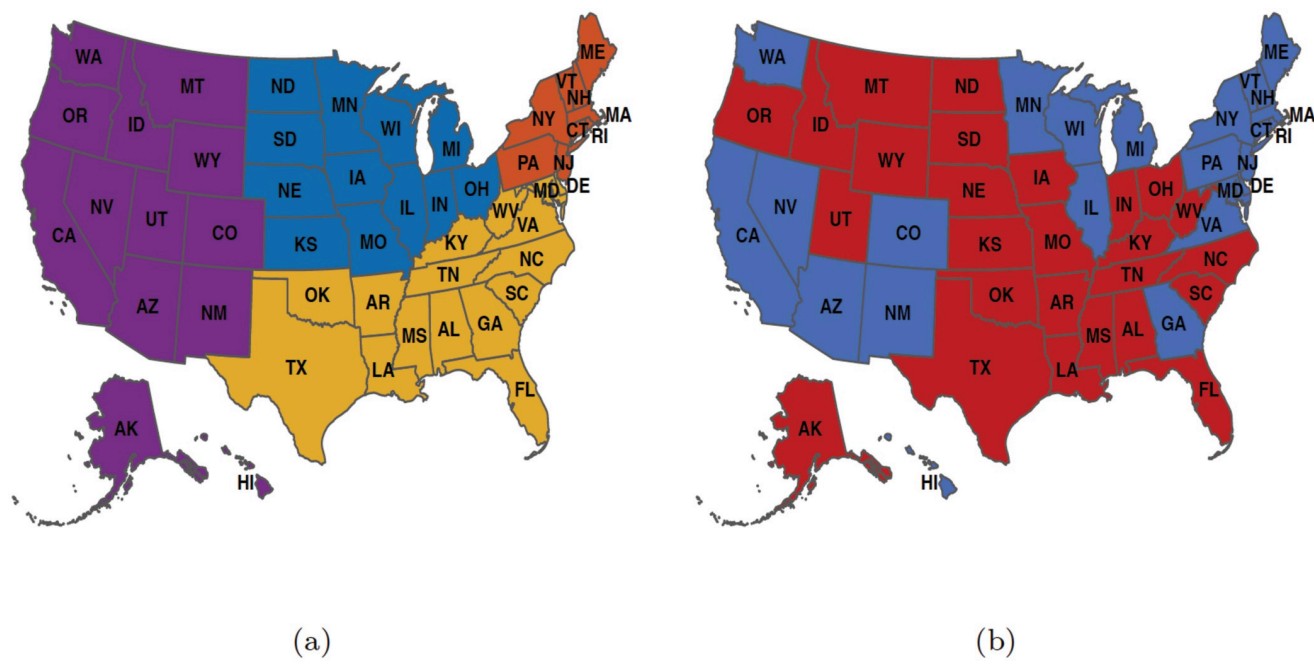

**Fig 1. Regional and political maps of the United States.** (a) A regional map where blue represents the midwest, orange represents the northeast, yellow represents the south, and purple represents the west; (b) A Political map where blue represents Democratic and red represents Republican party preference. The abbreviations are those given in the International Society for Organization Standard(ISO) for the United States [28]. The maps are generated using the R package usmap developed by Paolo Di Lorenzo, under a GNU General Public License (Version 3) [29].

We also grouped the data of the mental health indicators obtained from the survey data along geographical regions (see Fig 2) and political preference (see S1 Fig).

In order to see if there is any relationship between trends in mental health indicators and the trends in COVID-19 cases, hospitalization, and deaths, we equally grouped the COVID-19 related data from March 2nd, 2021 to January 10th, 2022 along regional (see Fig 3) and political preferences (see S2 Fig). From the COVID-19 related data we observed three distinguishable waves across the country. The first wave was from April 1st, 2021, to July 1st, 2021; the second wave was from July 2nd, 2021, to November 11th, 2021; and the third wave was from November 12th, 2021, to January 10th, 2022.

**Timeline of COVID-19 policies and mandates.** At the beginning of the pandemic, several policies were put in place to curtail the COVID-19 outbreak and as the pandemic progressed several mandates were enacted to alleviate the public's burden of the pandemic. Below in Tables 1 and 2 we list a timeline of a few policies implemented in the country. Some listed policies reflect efforts implemented to tackle the outbreak, such as a nationwide vaccine rollout for all adults, unemployment benefits, and an eviction moratorium. Depending on each policy's target population and topic, we postulate that some of these policies would impact individuals' mental health. The policies were categorized as related to anxiety and depression, or worried about one's finances.

## Clustering and correlation networks

To create the network across the states, we started by obtaining a correlation coefficient matrix for each pair of states using the time series data for each of the mental health-related indicators. For each indicator, we obtained a Pearson's correlation coefficient matrix, $\mathbb{P}_{x,y}$ using the

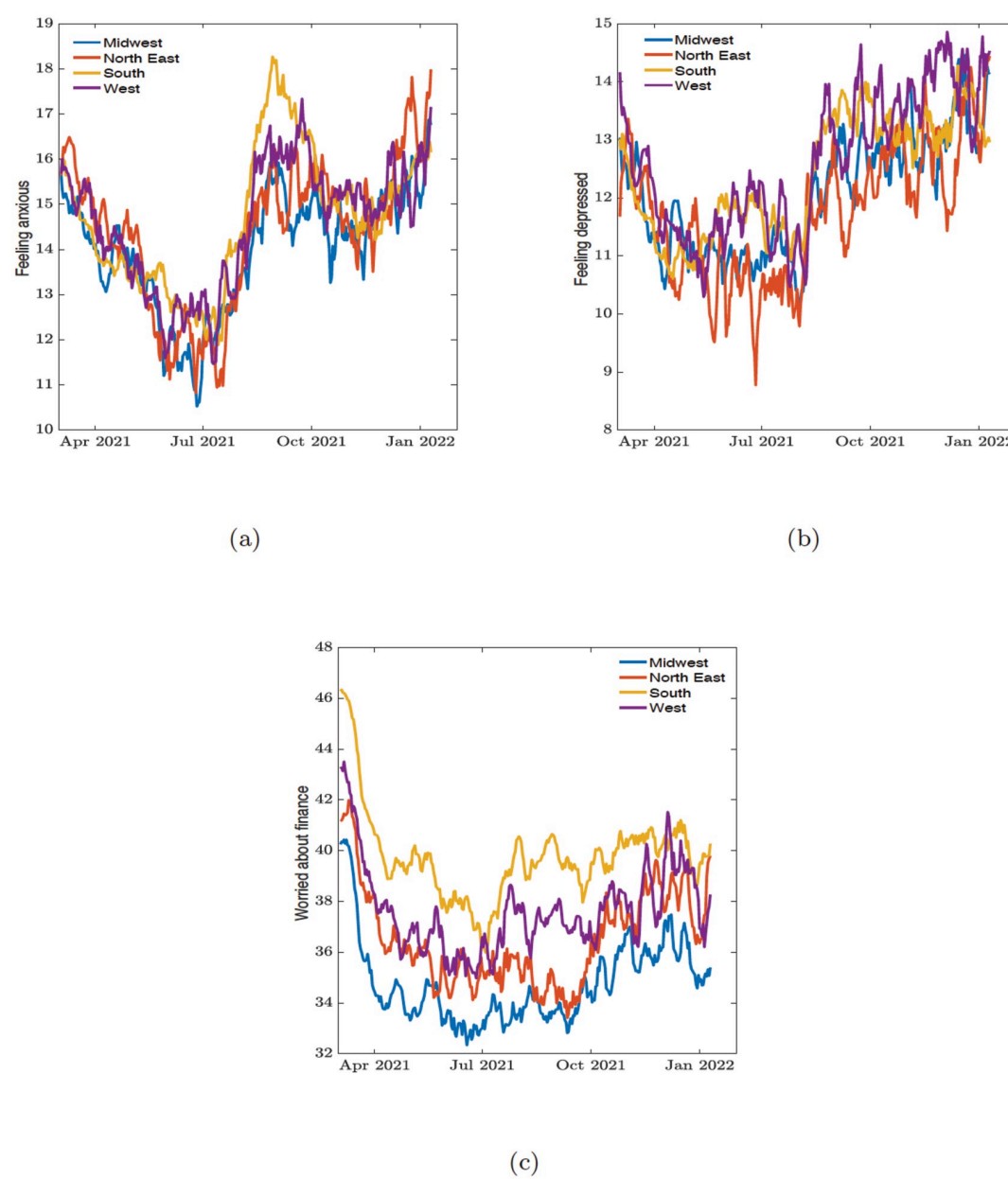

**Fig 2. Percentage of individuals.** (a) feeling anxious, (b) feeling depressed, and (c) worried about finances across demographic regions.

formula below [30].

$$\mathbb{P}_{x,y} = \frac{\sum_{i=1}^{N}(x_i - \bar{x})(y_i - \bar{y})}{\sqrt{\sum_{i=1}^{N}(x_i - \bar{x})^2 \sum_{i=1}^{N}(y_i - \bar{y})^2}},$$

where $x$ and $y$ present any two states in the country. If $\mathbb{P}_{x,y} \neq 0$, then the states $x$ and $y$ are connected. Otherwise, they are disconnected. Moreover, $\mathbb{P}_{x,y} > 0$ indicates a positive correlation

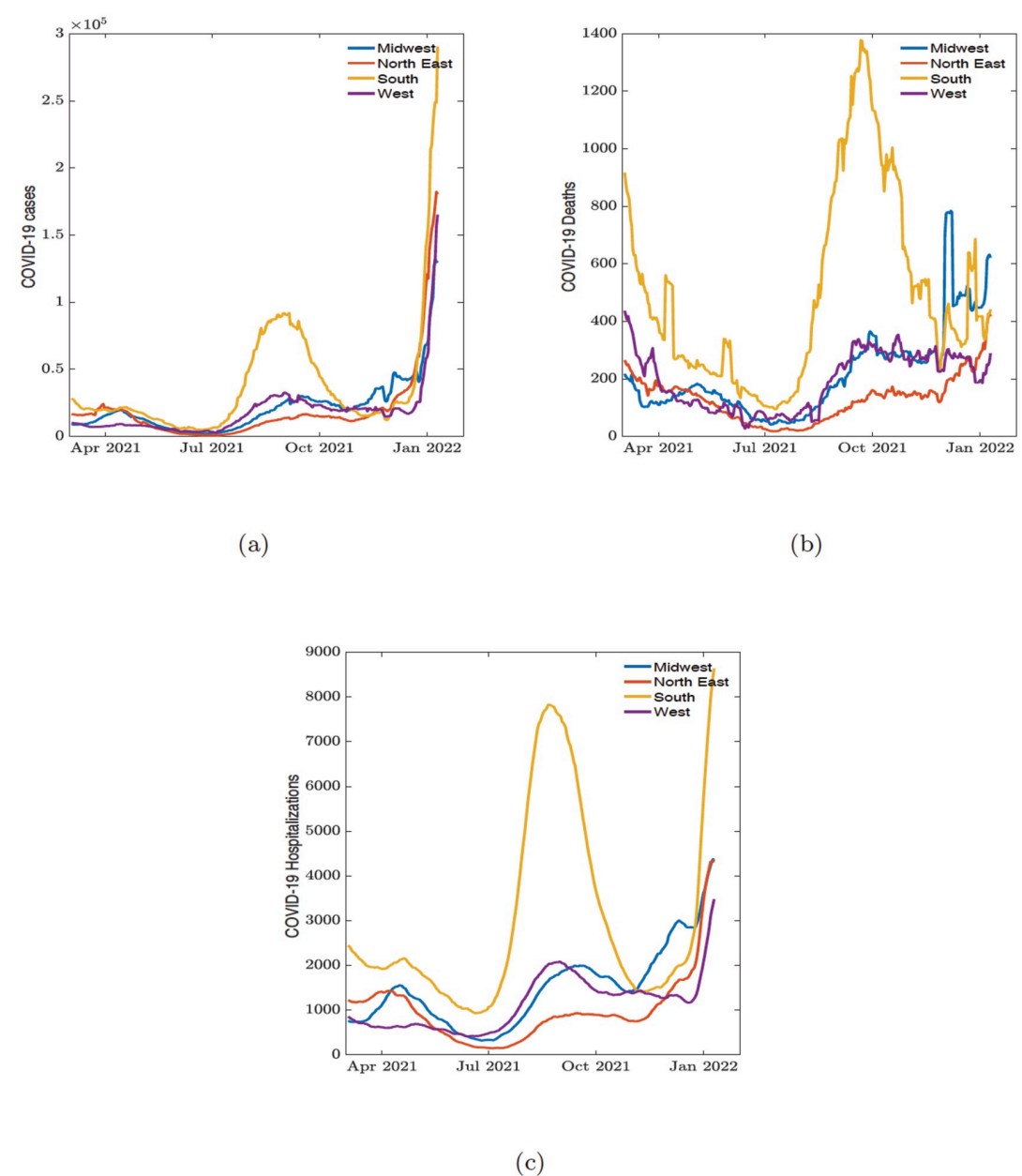

**Fig 3. Daily COVID-19.** (a) cases, (b) reported deaths, and (c) hospitalization cases across demographic regions.

and hence a positive connection between the states while $\mathbb{P}_{x,y} < 0$ indicates a negative correlation and a negative connection; the more positive (or negative) $\mathbb{P}_{x,y}$ is, the stronger the positive (or negative) correlation [30]. We used this correlation coefficient matrix to form the adjacency matrix, which defines the connectivity of states on a network with respect to a particular indicator. We used all the correlation coefficients unlike what is normally done where a cut-off is set and only strong associations are considered. This enabled us to incorporate the broad ranges of the states' similarities not only the positive associations.

**Table 1. Timeline of COVID-19 policies and mandates that affect feeling anxious and depressed.**

| Date | Policies and mandates |
| --- | --- |
| April 19, 2021 | All adult Americans eligible for vaccines by 4.19 |
| May 13, 2021 | CDC guidance on masking updated, no longer recommends masks indoors or outdoors for fully vaccinated people regardless of the number of people gathered |
| July 03, 2021 | CDC states more than 50% of cases are Delta variant |
| October 20, 2021 | President Biden announces plan for distributing vaccine to children 5—11 FDA authorizes Moderna and J&J vaccine in eligible populations, and allows mixing vaccine doses (can use different primary and booster shots) |
| December 09, 2021 | FDA expands Pfizer-BioNTech booster eligibility to 16- and 17-year olds |
| December 27, 2021 | CDC shortens isolation and quarantine period |

**Table 2. Timeline of COVID-19 policies and mandates that affect finances.**

| Date | Policies and mandates |
| --- | --- |
| March 11, 2021 | President Biden signs a coronavirus relief package bill including extended unemployment benefits and a third stimulus check |
| April 21, 2021 | Tax credit for small businesses and non-profits, covers 80h of leave, $511/day |
| July 31, 2021 | Moratorium on eviction expires |
| August 3, 2021 | CDC issues eviction moratorium specifically in areas with 'substantial or high' rates of COVID transmission |
| August 26, 2021 | Moratorium ends by Supreme Court ruling |
| September 06, 2021 | Unemployment benefits end for many individuals |

Once the connectivity network is obtained, we proceed to find cluster(s) on the network. A cluster consists of states (or nodes) that are more similar (with respect to a particular indicator) to each other than to states in other clusters. Whether a given pair of states are grouped together depends on both how they are connected to each other and how they are connected to other states in the network. We utilized Traag's implementation of the Leiden algorithm [31] in Python 3.9.7 from Spyder version 5.2.2 (specifically, the find_partition and optimise_partition_multiplex functions from the leidenalg package [32]). Briefly, these functions were used as follows: Each network was split into a positive and negative sub-network. The positive sub-network contained only the positive connections from the original network, and the negative sub-network contained only the negative connections from the original network [32]. We applied the find_partition function to both sub-networks to obtain clusters for each, and we thereafter applied the optimise_partition_multiplex_function to both sets of clusters to obtain a single set of clusters for the original network.

The clusters returned by Leiden's algorithm are not unique, meaning that if Leiden's algorithm is run multiple times, the clusters may not be the same. Consensus clustering addresses this issue by grouping states based on multiple runs of the Leiden algorithm [33]. Specifically, the Leiden algorithm is run multiple times with a given network (hereafter referred to as the original network), and after all runs are completed, the frequencies at which states are grouped together are used to create a new adjacency matrix (and thus a new network); the Leiden algorithm is then run with this new network. The resulting clusters are referred to as consensus clusters because they represent the "consensus" of multiple Leiden algorithm runs with the

original network. The number of runs necessary may be assessed by tracking how the above-mentioned frequencies change as more runs are performed. We manually implemented consensus clustering in Python with 300 runs of the Leiden algorithm [32]. We verified that 300 runs were sufficient by visually inspecting how frequencies changed with more runs. The results from the final iteration for each indicator consensus matrix were stored, then mapped onto United States maps to visually assess the memberships for each indicator.

Fig 4 shows the membership results of the consensus clustering for the three mental health indicators. We noticed a south geographical region clustering for both the feeling anxious and worried about finances indicators in Fig 4(a) and 4(c) while Fig 4(b) shows no distinct clustering pattern, indicating that the communities with similar trends for the feeling depressed indicator did not cluster in manners reflecting geographical regions or political party affiliation shown in Fig 1(a) and 1(b).

The membership results obtained from the final iteration of the consensus clustering across the three mental health indicators were used to construct an allegiance matrix. The allegiance matrix gives "the probability for two regions being in the same community across all COVID-19 mental health indicators" [19, 34]. The map of the allegiance matrix is shown in Fig 5(a) and the corresponding heat map is shown in Fig 5(b). These indicate which states were most likely to be grouped together (or exhibit similar trends) across all three mental health indicators. The map shows three main clusters, of which the light blue colored states are the most interesting. The clustering of the light blue states closely looks like the south geographic region, with the exception of Virginia, West Virginia, and Arkansas, which were not as frequently grouped together with the rest of the south. The light blue colored states also include California, North Carolina, and Nevada, which are non-southern states.

## Dynamic connectome analysis

We follow the approach by Amico and Bulai [19], and use the sliding window analysis to compute the dynamic connectome for each mental health indicator. The sliding window analysis is a technique constructed by taking a fixed time interval to explore the different indicators at various time frames. In our sliding window analysis, we take a fixed window of 30 days and slide this by 1 day and we computed the average correlation values for our mental health indicator across the network within this window. Each data point is thus a correlation value taken across the 30 days within a window. These slides or snapshots lead to a time series, which we plotted to display the dynamic fluctuations in the indicator of interest across the time frame used.

For each of the mental health indicators, we computed also the eigenvector centrality of the corresponding network and the related dynamic connectome. The eigenvector centrality measures the transitive influence of each state. In other words, it gives information about the importance of a state (node). Connections originating from high-scoring states contribute more to the eigenvector score of a state than connections from low-scoring states. A state with a high eigenvector score means that the state is connected to many states that have high scores [35].

## Results

In this section we present the results of the dynamic connectome for the correlation network (see Figs 6 and 8) and the related eigenvector centrality values (see Figs 7 and 9) along political party preference and demographic regional lines. All analyses were conducted in R 2021 version and Python 3.9.7 from Spyder version 5.2.2, while the figures were plotted in MATLAB R2021b.

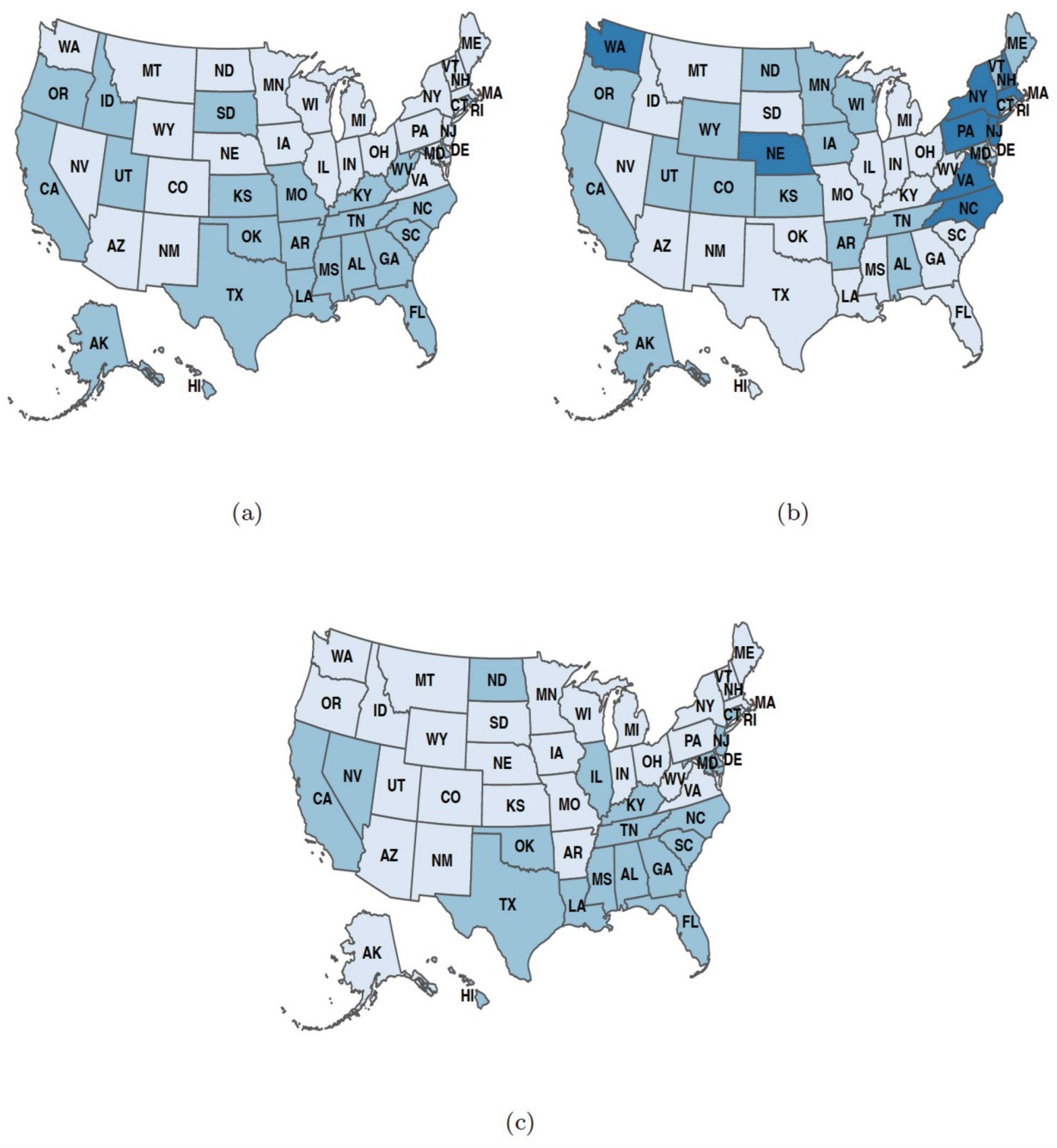

**Fig 4.** Country-wide maps grouping states with similar trends for (a) feeling anxious, (b) feeling depressed, and (c) worried about finances indicators. The states with light blue, blue, and dark blue each represent distinct clusters. The maps are generated using the R package usmap developed by Paolo Di Lorenzo, under a GNU General Public License (Version 3) [29].

As mentioned earlier, we observed three waves in the COVID-19 related data during time frame used. This split the dynamic connectome of each indicator into three. We indicated this with the vertical dash lines, the horizontal dash line shows the width of the wave. For each of the Figs 6–9, we record the maximum and minimum correlation values within each wave in Tables 3 and 5. We also record the maximum and minimum eigenvector centrality values (see

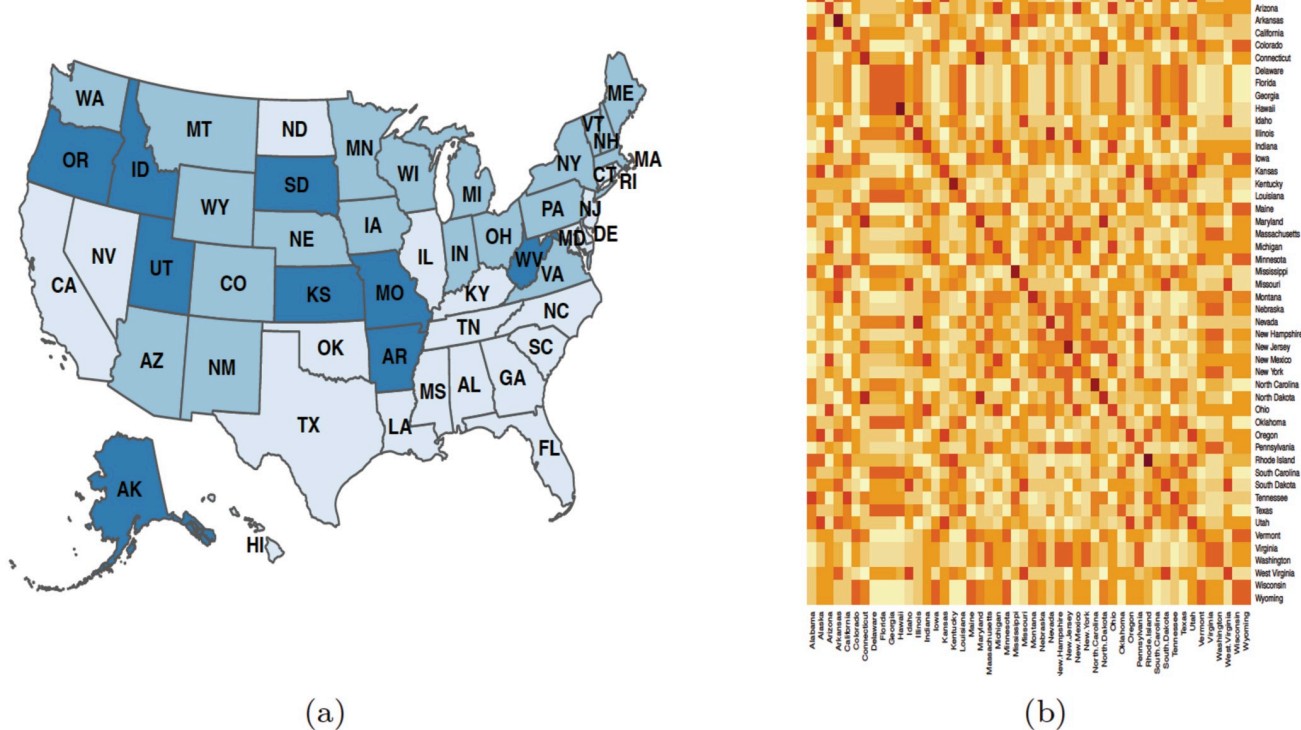

**Fig 5. Allegiance clustering for feeling anxious, feeling depressed, and worried about finances indicators.** (a) Allegiance map; (b) Allegiance Heat map. The states with the same color (light blue, blue, or dark blue) are in the same cluster. The map is generated using the R package usmap developed by Paolo Di Lorenzo, under a GNU General Public License (Version 3) [29].

Tables 4 and 6). The eigenvector centrality highlights regions or parties which are most important, especially in times of lower correlation.

To minimize confusion when referencing the dynamic connectome results of each of the three indicators, we denote all future references by use of the term "variable" rather than "indicator". Changes in the collected survey data will be discussed in the results and discussion sections and refer to as indicators, while reference to the results of the dynamic connectome (such as the time series changes in the correlation and eigenvector centrality values) will be refered to as variable.

### Regional dynamic connectome

For the regional dynamic connectome results of the feeling anxious variable, both the lowest minimum (-0.0648) and highest maximum (0.4523) correlation values within the first wave were observed from the northeastern region (see Fig 6(a) and Table 3). During the second wave, the minimum correlation value (-0.0762) remained in the northeast, and the maximum correlation value (0.5753) was recorded from the south. The third wave period showed the minimum correlation value (-0.0694) in the West, and the maximum correlation was seen in the midwest region (0.0836).

For the correlation values in the feeling depressed variable, the first wave had both maximum (0.5958) and minimum (-0.0690) correlation values in the midwest region (see Fig 6(b) and Table 3). The second wave period exhibited a similar pattern to the feeling anxious variable, with the northeast having the minimum correlation (-0.0867) and the south with the

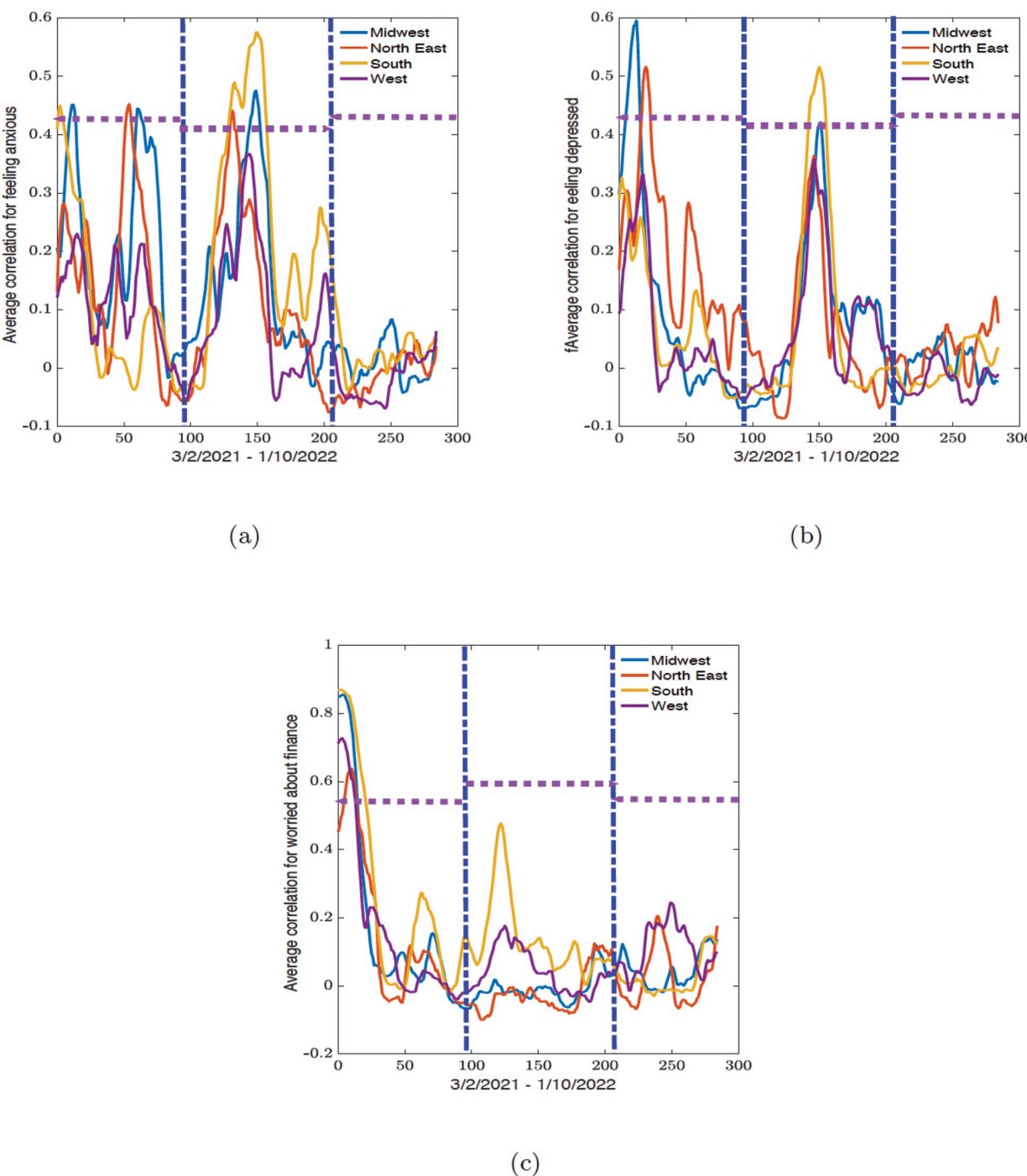

(a)

(b)

(c)

**Fig 6.** Time series of the average correlation for states in midwest, northeast, southern, and western regions obtained from the sliding window analysis for (a) feeling anxious, (b) feeling depressed, and (c) worried about finances variables. The vertical blue dot-dash lines indicate the start and end of the epidemic waves observed in Fig 3 and the horizontal dash lines indicate the width of the waves.

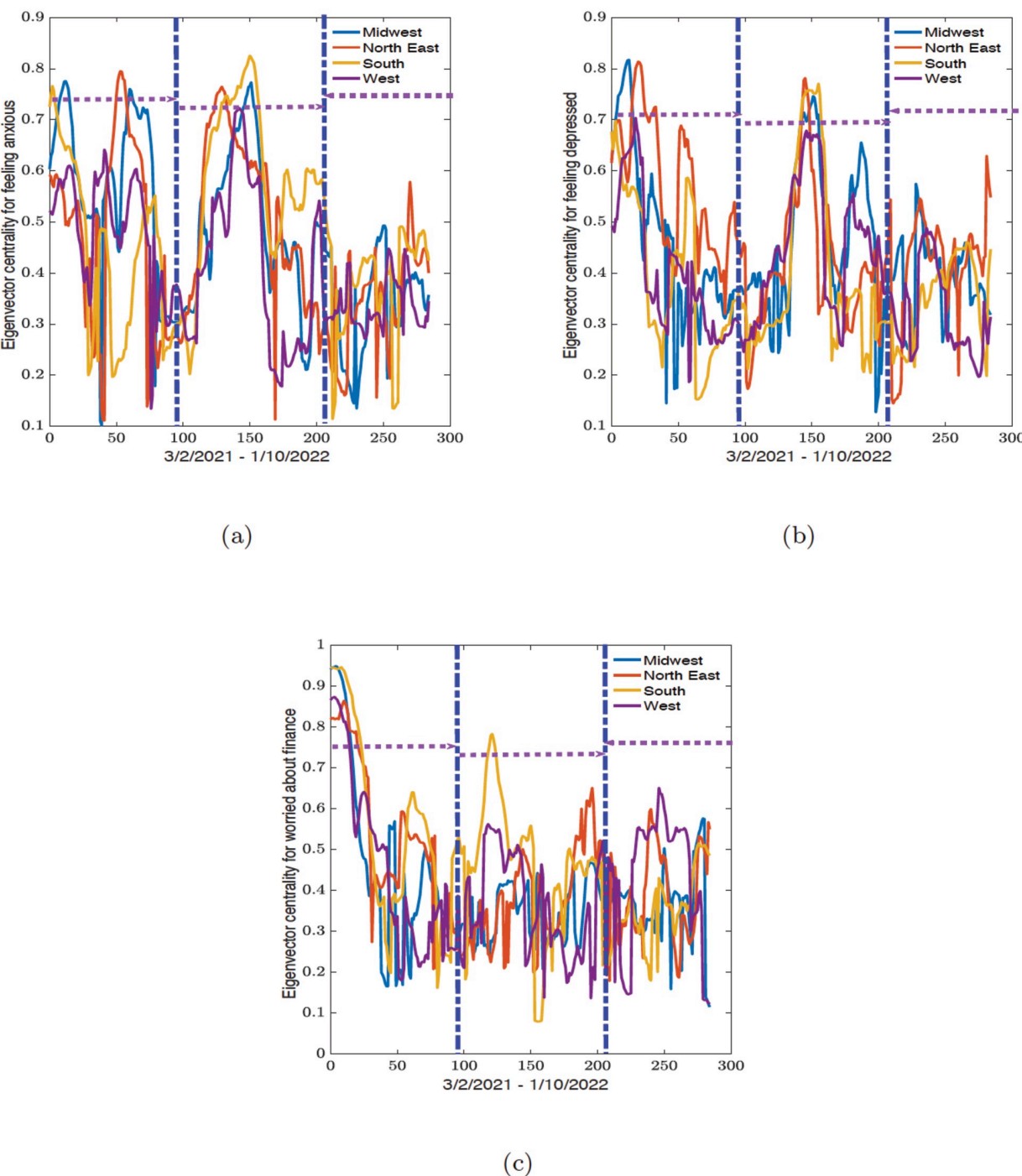

**Fig 7.** Time series of the average eigenvector centrality for states in midwest, northeast, southern, and western regions obtained from the sliding window analysis for (a) feeling anxious, (b) feeling depressed, and (c) worried about finances variables. The vertical blue dot-dash lines indicate the start and end of the epidemic waves observed in Fig 3 and the horizontal dash lines indicate the width of the waves.

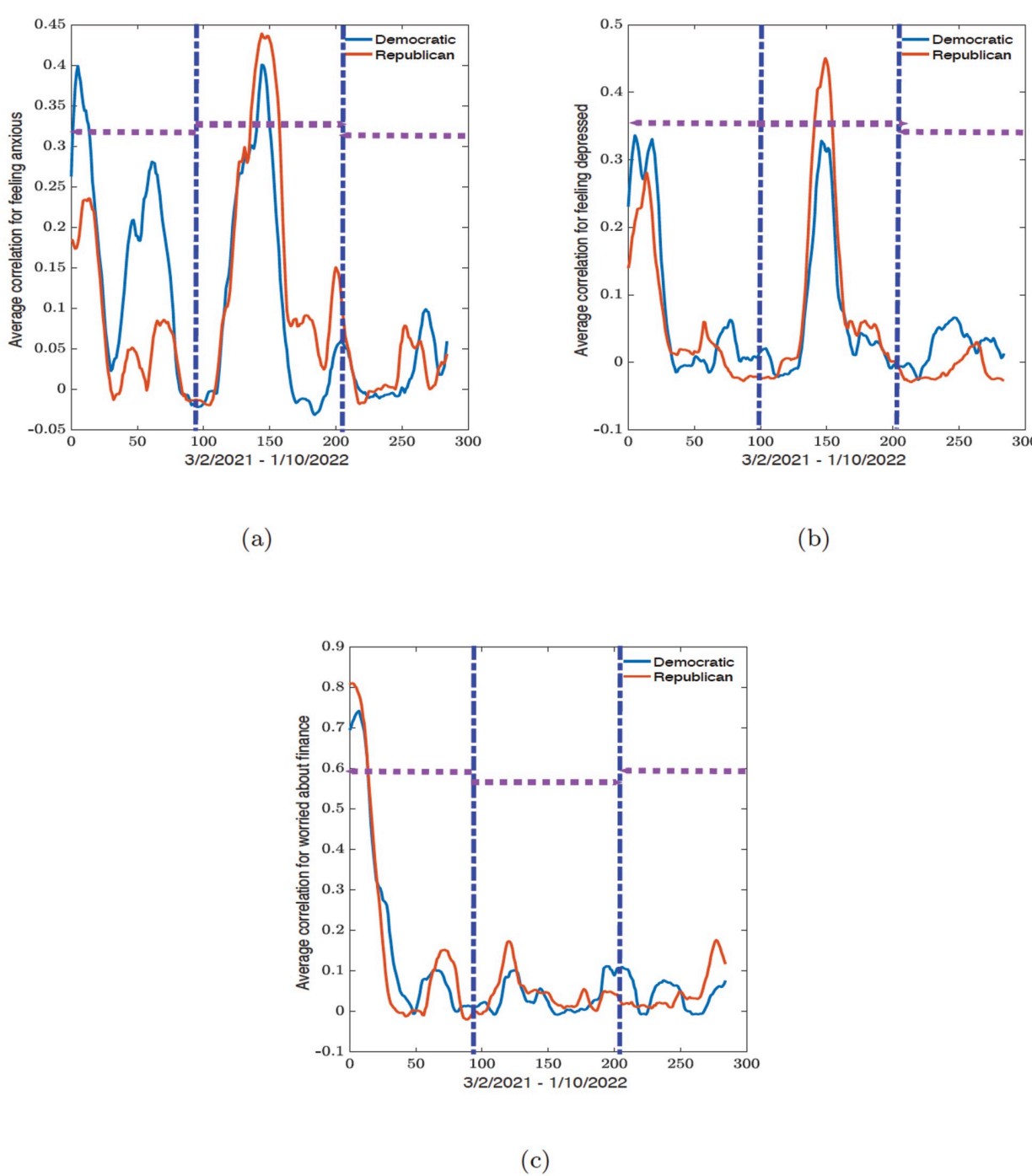

**Fig 8.** Time series of the average correlation within Democratic and Republican states obtained from the sliding window analysis for (a) feeling anxious, (b) feeling depressed, and (c) worried about finances variables. The vertical blue dot-dash lines indicate the start and end of the epidemic waves observed in Fig 3 and the horizontal dash lines indicate the width of the waves.

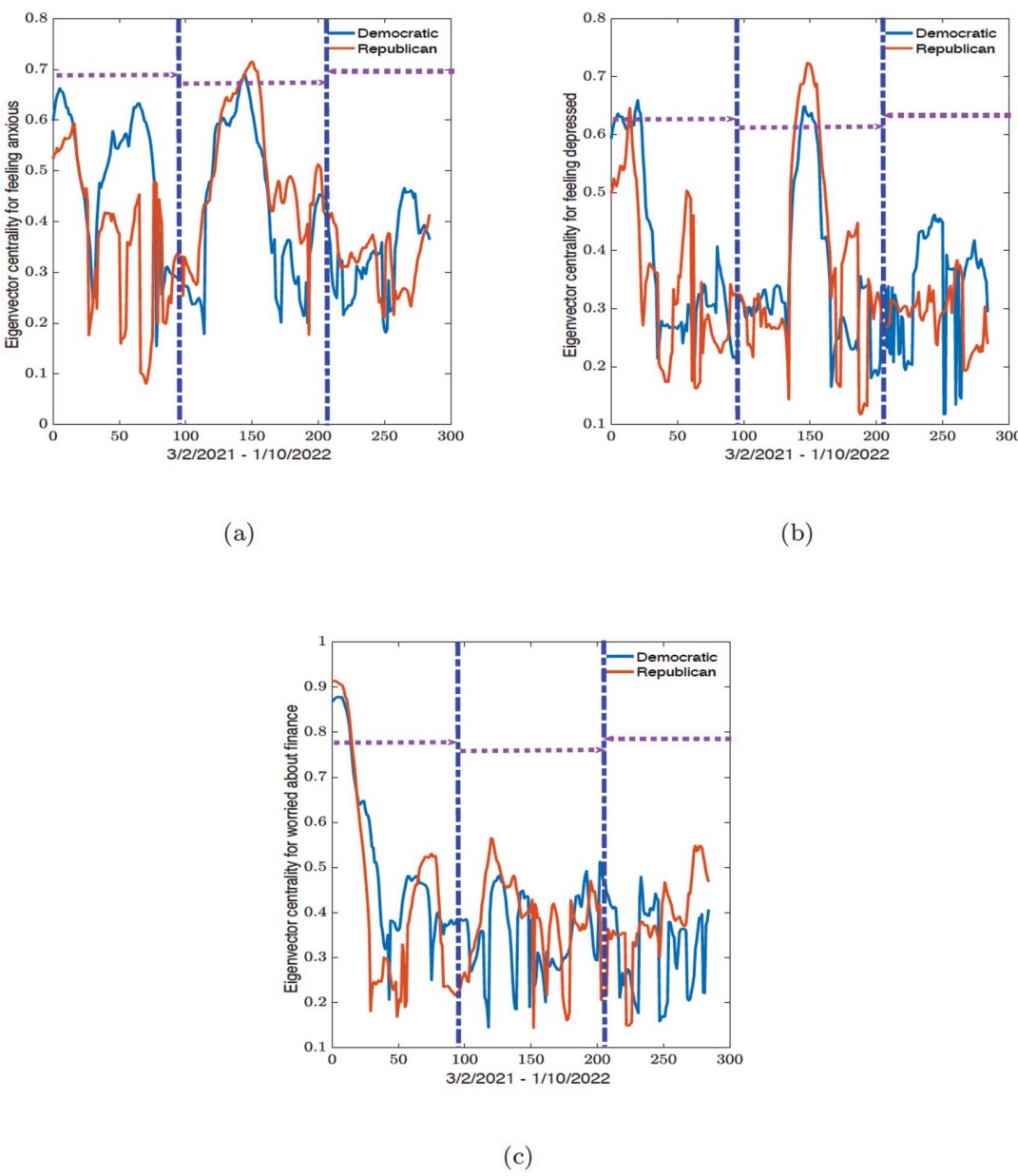

**Fig 9.** Time series of the average eigenvalue centrality for Democratic and Republican states obtained from the sliding window analysis for (a) feeling anxious, (b) feeling depressed, and (c) worried about finances variables. The vertical blue dot-dash lines indicate the start and end of the epidemic waves observed in Fig 3 and the horizontal dash lines indicate the width of the waves.

maximum correlation (0.5160). The third wave period continued with the west having the minimum correlation (-0.0633) among all four regions and the northeast with the maximum correlation (0.1222).

The remaining variable, worried about finances in Fig 6(c) and Table 3, exhibited higher maximum correlation values during the first wave, which decreased with each wave. During the first wave, the midwest had the minimum correlation value (-0.0565), and the south had the maximum correlation value (0.8694). The south region maintained the highest correlation

**Table 3. Minimum and maximum values of correlation time series for feeling anxious, feeling depressed, worried about finances between demographic regions.**

| Regions | Wave | Anxious | Anxious | Depressed | Depressed | Finance | Finance |
|---------|------|---------|---------|-----------|-----------|---------|---------|
|  |  | Min | Max | Min | Max | Min | Max |
| Midwest | 1 | -0.0171 | 0.4514 | -0.0690 | 0.5958 | -0.0565 | 0.8554 |
| Midwest | 2 | -0.0317 | 0.4755 | -0.0689 | 0.4213 | -0.0668 | 0.1232 |
| Midwest | 3 | -0.0426 | 0.0836 | -0.0291 | 0.0614 | -0.0186 | 0.1428 |
| North East | 1 | -0.0648 | 0.4523 | -0.0033 | 0.5166 | -0.0524 | 0.6376 |
| North East | 2 | -0.0762 | 0.4413 | -0.0867 | 0.3642 | -0.1000 | 0.1245 |
| North East | 3 | -0.0677 | 0.0478 | -0.0330 | 0.1222 | -0.0688 | 0.2052 |
| South | 1 | -0.0421 | 0.4498 | -0.0525 | 0.3267 | -0.0216 | 0.8694 |
| South | 2 | -0.0365 | 0.5753 | -0.0453 | 0.5160 | 0.0026 | 0.4765 |
| South | 3 | -0.0382 | 0.0609 | -0.0464 | 0.0583 | -0.0301 | 0.1457 |
| West | 1 | -0.0212 | 0.2302 | -0.0524 | 0.3317 | -0.0396 | 0.7274 |
| West | 2 | -0.0559 | 0.3675 | -0.0542 | 0.3576 | -0.0472 | 0.1770 |
| West | 3 | -0.0694 | 0.0633 | -0.0633 | 0.0291 | -0.0143 | 0.2447 |

values (0.4765) during the second wave, and the minimum value (-0.1000) was in the northeast. For the third wave, the northeast still had the lowest correlation value (-0.0688) within the four regions, and the west had the maximum value (0.2447).

For all three variables across each wave, the recorded minimum values were very weak negative correlations (all under -0.1); the states could be considered to have no correlation within each region. The correlation values for the first and second waves were mostly moderate to strong correlation values (between 0.3576 to 0.5753) unlike the correlation values for the third wave which had weak positive correlation values (between 0.0291 to 0.2447). This excludes the worried about finances variable, which the second wave has a weak positive correlation (between 0.1232 to 0.177, except for the southern states (with a medium positive correlation, 0.4765)). The outcomes for the largest minimum and maximum correlation values for the second wave were the same for all three variables, with the minimum being the northeast, and the maximum being the south.

Fig 7 and Table 4 show the results of the dynamic connectome of the eigenvector centrality values for all the mental health variables. We observed that during the first wave for the feeling anxious variable, the northeast had the highest maximum value (0.7954) while the midwest had the smallest minimum eigenvector centrality value (0.1014). The minimum and maximum values during the second wave were recorded in the northeast and south (0.1128 and 0.8252, respectively). This position flips during the third wave, where the south now had the lowest minimum value (0.1344) and the northeast has the largest maximum (0.5786).

For the dynamic connectome results for the feeling depressed variable, only two regions are recorded for the largest maximum and lowest minimum eigenvector centrality values. The midwest had the smallest minimum value for the first and second waves (0.1446 and 0.1273, respectively) and the maximum value of 0.8178 during the first wave. The northeast had the largest maximum values in the second (0.7820) and third (0.6298) waves, and the minimum correlation value of 0.1622 during the third wave.

For the worried about finances variable, the south had the minimum eigenvector centrality value of 0.1617 during the first wave while the midwest had the maximum eigenvector centrality value of 0.9493. During the second wave, the south recorded both the lowest minimum and maximum eigenvector centrality values of 0.0791 and 0.7831, respectively. The midwest and western states had the minimum eigenvector centrality value of 0.1148 and a maximum value of 0.6512 during the third wave.

**Table 4. Minimum and maximum values of eigenvector centrality time series for feeling anxious, feeling depressed, worried about finances between demographic regions.**

| Regions | Wave | Anxious | Anxious | Depressed | Depressed | Finance | Finance |
|---|---|---|---|---|---|---|---|
|  |  | Min | Max | Min | Max | Min | Max |
| Midwest | 1 | 0.1014 | 0.7760 | 0.1446 | 0.8178 | 0.1654 | 0.9493 |
| Midwest | 2 | 0.2104 | 0.7737 | 0.1273 | 0.7464 | 0.2076 | 0.4684 |
| Midwest | 3 | 0.1345 | 0.4932 | 0.2303 | 0.5749 | 0.1148 | 0.5763 |
| North East | 1 | 0.1111 | 0.7954 | 0.3198 | 0.8142 | 0.2076 | 0.8642 |
| North East | 2 | 0.1128 | 0.7650 | 0.1445 | 0.7820 | 0.1613 | 0.6512 |
| North East | 3 | 0.1557 | 0.5786 | 0.1622 | 0.6298 | 0.1871 | 0.5992 |
| South | 1 | 0.1966 | 0.7666 | 0.1523 | 0.6981 | 0.1617 | 0.9465 |
| South | 2 | 0.1141 | 0.8252 | 0.2042 | 0.7711 | 0.0791 | 0.7831 |
| South | 3 | 0.1344 | 0.4915 | 0.1982 | 0.4606 | 0.1799 | 0.5155 |
| West | 1 | 0.1340 | 0.6420 | 0.1863 | 0.7038 | 0.1804 | 0.8739 |
| West | 2 | 0.1778 | 0.7232 | 0.2450 | 0.6793 | 0.1360 | 0.5625 |
| West | 3 | 0.2715 | 0.4480 | 0.1966 | 0.4882 | 0.1225 | 0.6512 |

To summarize, during the first wave, the midwest and northeast have the highest average correlation for feeling anxious and feeling depressed. However, with worried about finances, we have a weak correlation between each region except during the initial part of the first wave where the southern states have the highest average correlation values. During the second wave, we see that the southern states consistently had the highest average correlation values for feeling anxious, feeling depressed, and worried about finances. Lastly, during the third wave, there were no distinctive patterns for all the indicators, and the overall correlation is weak and even negative for some indicators. This simply shows that during this period in time in the pandemic there are no strong similarities between the states.

Lastly, for the eigenvector centrality, we observed that the midwest and northeast have the largest eigenvector centrality values during the first wave while the southern states had the largest eigenvector centrality value during the second wave. There was also no distinctive pattern during the third wave, but the values are much higher than the average correlation values.

### Political dynamic connectome

For political party preferences, the dynamic connectome for the correlation and eigenvector centrality are plotted in Figs 8 and 9. The maximum and minimum values for all three variables are summarized in Tables 5 and 6.

**Table 5. Minimum and maximum values of correlation time series for feeling anxious, feeling depressed, worried about finances between political parties.**

| Parties | Wave | Anxious | Anxious | Depressed | Depressed | Finance | Finance |
|---|---|---|---|---|---|---|---|
|  |  | Min | Max | Min | Max | Min | Max |
| Democratic | 1 | -0.0196 | 0.3994 | -0.0151 | 0.3365 | -0.0061 | 0.7423 |
| Democratic | 2 | -0.0316 | 0.4007 | -0.0217 | 0.3284 | -0.0081 | 0.1117 |
| Democratic | 3 | -0.0111 | 0.0991 | -0.0258 | 0.0663 | -0.0077 | 0.0770 |
| Republican | 1 | -0.0145 | 0.2358 | -0.0277 | 0.2810 | -0.0199 | 0.8104 |
| Republican | 2 | -0.0196 | 0.4394 | -0.0293 | 0.4507 | -0.0085 | 0.1735 |
| Republican | 3 | -0.0181 | 0.0787 | -0.0277 | 0.0301 | 0.0077 | 0.1764 |

**Table 6. Minimum and maximum values of eigenvalue centrality time series for feeling anxious, feeling depressed, and worried about finances between political parties.**

| Parties | Range | Anxious | Anxious | Depressed | Depressed | Finance | Finance |
|---|---|---|---|---|---|---|---|
|  |  | Min | Max | Min | Max | Min | Max |
| Democratic | 1 | 0.1539 | 0.6633 | 0.2133 | 0.6597 | 0.2062 | 0.8785 |
| Democratic | 2 | 0.1781 | 0.6905 | 0.1651 | 0.6492 | 0.1448 | 0.5127 |
| Democratic | 3 | 0.1810 | 0.4663 | 0.1176 | 0.4621 | 0.1585 | 0.4794 |
| Republican | 1 | 0.0804 | 0.5932 | 0.1624 | 0.6461 | 0.1684 | 0.9139 |
| Republican | 2 | 0.1764 | 0.7153 | 0.1173 | 0.7237 | 0.1437 | 0.5652 |
| Republican | 3 | 0.2116 | 0.4143 | 0.1931 | 0.3836 | 0.1490 | 0.5483 |

From Table 5, we observed that the lowest minimum and largest maximum correlation values (-0.0196 and 0.3994) during the first wave for the feeling anxious variable was recorded for the Democratic states. During the second wave, the minimum correlation value (-0.0316) remained in the Democratic states while the highest maximum correlation value (0.4394) was among the Republican states. In the third wave, these positions are flipped, where the largest minimum value (-0.0181) was among the Republican states and the maximum within the Democratic states (0.0991).

For the feeling depressed variable, the largest maximum correlation (0.3365 and 0.0663) was observed during the first and third waves in the Democratic states, while the largest maximum correlation value was observed in the Republican states for the second wave (0.4507). The Republican states have the largest minimum correlation value across all three waves (-0.0277, -0.0293, and -0.0277 respectively).

For the worried about finances variable, all minimum and maximum correlation values are largest among the Republican states. The largest minimum values for the three waves are (-0.0199, -0.0085, and -0.0077), and the largest maximum correlation values are (0.8104, 0.1735, and 0.1764).

Fig 9 and Table 6 show the results of the eigenvector centrality values for all three mental health variables. Looking at the minimum and maximum values for the eigenvector centrality for the feeling anxious variable in Table 6 we observed that the Republican states had the minimum value of 0.0804 during the first wave, while Democratic states had the maximum eigenvector centrality values of 0.6633. During the second wave, the minimum and maximum values (0.1764 and 0.7153) were recorded for the Republican states. In the third wave, the minimum and maximum eigenvector centrality values (0.1810 and 0.4663) were seen within the Democratic states.

For the eigenvector centrality values for the feeling depressed variable, the Republican states had the minimum values in the first and second waves (0.1624 and 0.1173) and Democratic states (0.1176) during the third wave. The maximum values during the first and third waves were observed among the Democratic states as (0.6597) and (0.4621). The Republican states had the maximum eigenvector centrality value of 0.7237 during the second wave.

The Republican states had the minimum eigenvector centrality values of 0.0804 and 0.1764 during the first and second waves for the feeling anxious variable while the Democratic states had the minimum value of 0.1810 during the third wave. The maximum eigenvector centrality values in the first and third waves are 0.6633 and 0.4663 among the Democratic states while the Republican states had a maximum eigenvector centrality value of 0.7153 during the second wave.

When observing the eigenvector centrality values for the feeling depressed variable, the lowest minimum and largest maximum values have a similar party preference outcome as the eigenvector centrality values for the feeling anxious variable; Republican states had the minimum values in the first and second wave (0.1624 and 0.1173) and Democratic states (0.1176) during the third wave. A similar party outcome as feeling anxious was observed for the maximum eigenvector centrality values, namely Democratic states in the first and third wave (0.6597 and 0.4621) and Republican in the second wave (0.7237).

In the political party preference dynamic connectome results for worried about finances, all the lowest minimum and largest maximum eigenvector centrality values are among the Republican states. The lowest minimum values for the three waves are (0.1684, 0.1437, and 0.1490), and the largest maximum eigenvector centrality values are (0.9139, 0.5652, and 0.5483). Across the eigenvector centrality values for all three variables, the Republican states consistently recorded the highest maximum values during the second wave.

To summarize, during the first wave, the Democratic states had the highest average correlation for feeling anxious and feeling depressed. However, with worried about finances, we have a weak correlation between each region or party except during the initial part of the first wave where the Republican states have the highest average correlation. During the second wave, we see that the Republican states consistently have the highest average correlation values for feeling anxious, feeling depressed, and worried about finances. During the third wave, there are no distinctive patterns for all the indicators, and the overall correlation is weak and even negative for some indicators. This simply shows that during this period of time in the pandemic, there are no strong similarities between the states.

For the dynamic connectome for the eigenvector centrality across regional and political party preference values. We also see similar trends with the Democrats having the largest eigenvector centrality values during the first wave. While the Republican states have the largest eigenvector centrality value during the second wave. There are also no distinctive patterns during the third wave, but the values are much higher than the average correlation values.

## Discussion

In this study, we applied clustering analysis from network theory to understand the connectivity and similarities between states and the impacts of COVID-19 on mental health across the country using COVID-19 related mental health indicators such as worried about finances, feeling depressed, and feeling anxious. We framed our results in terms of three time periods of three COVID-19 waves identified between March 2nd, 2021 to January 10th, 2022 using daily COVID-19 related cases, deaths, and hospitalizations (see Fig 3).

At the beginning and as the pandemic progressed several policies and mandates were put in place to curtail the outbreak; and several mandates were enacted to alleviate the public's burden of the pandemic, see Tables 1 and 2 for a list of some of the implemented policies and mandates. To assess the impact of these policies on people's mental health, we follow the approach by Amico and Bulai [19] and superimpose the network correlation and similarity outcomes onto the COVID-19 related data and identify the policy implemented during these times. This allowed us to see if a correlation value was indicative of the COVID-19 cases, hospitalization, and deaths and implemented policy. Note we have used consensus clustering to group similar states together and dynamic connectome from the sliding window analysis to create a time series of their relationship.

We discuss the outcome from the political party preference dynamic connectomes (see Fig 8) since there are fewer curves on the plots, unlike the regional dynamic connectomes in Fig 6. The conclusions of the results for the geographical regions are the same as we show later.

## Policy impact on feeling anxious and depressed

We will start with feeling anxious and feeling depressed. For either of these mental health variables, we see in Fig 8(a) and 8(b) that the Democratic states have the highest correlation values during the first wave from March 2nd, 2021 to July 1st, 2021. During this period, there were two policies introduced: the first policy was on April 19th (the 19th-day mark on the connectome) that guaranteed all American adults are eligible for the COVID-19 vaccine. During this period, we observed a decrease in reported percentages of individuals feeling anxious and depressed. Thus, the introduction of a policy that would assist people in getting vaccinated and protecting themselves during the pandemic can reasonably be understood to work towards a decrease in reported percentages of individuals feeling anxious and depressed.

The second policy was a CDC guideline update on masking made on May 13 (the 43rd-day mark on the connectome). In their guideline, the CDC retracted their recommendation for fully vaccinated individuals to wear masks indoors or outdoors. This policy is, however, ambiguous as some individuals can consider this guideline to indicate a decrease in the severity and danger of the COVID-19 pandemic, while others may see this as a premature decision and worry about the potential outcomes of making this announcement.

During the second wave from July 2nd, 2021 to November 11th, 2021, we see an increase in the correlation values within the Republican states at the 165-day mark on the connectome. However, during this time, CDC announced on July 3rd that the COVID-19 Delta variant was responsible for over 50% of all COVID-19 cases. Simultaneously, the Republican states recorded a higher percentage of individuals feeling anxious and depressed. This observervation is reasonable given the large spike in COVID-19 cases, hospitalizations, and deaths seen in the Republican states.

During the third wave, from November 12th, 2021, to January 10th, 2022, the correlation between the states was weak. During this period, FDA expanded Pfizer-BioNTech booster eligibility to 16- and 17-year olds, and CDC shortens the isolation and quarantine period. Furthermore, during this period, COVID-19 cases, and hospitalization were on the rise mostly in the Democratic states, while the deaths were higher in the Republican states which explains higher percentage of individuals with anxiety and depression in the Republican states.

For the regional connectome for feeling anxious, and feeling depressed, we observed that during the first wave at the 40 and 80-day mark on the connectome (see Fig 6(a) and 6(b)), the midwest and the northeast had the highest average correlation values. The same factors influencing the dynamic connectome for the political party preference during this period also affect the regional connectome. During the second period, the highest correlation value was observed in the southern states at the 110-day mark on the connectome in Fig 6(a) and 6(b). Lastly, during the third wave, there was no clear correlation between the states.

## Policy impact on worries about finances

For the worried about finances indicator, we see in Fig 8(c) an initial high correlation value among both Republican and Democratic states during the first 30 days of the collected survey responses in the first wave period. This correlation aligns with a collective decrease in the percentage of people worried about their finances during the first few days of this time period (see Fig 2(c)). This decline may have been facilitated by the signing of the COVID-19 relief package by President Biden; this relief package extended unemployment compensation and included a third stimulus check.

Another possible explanation for the decreasing trend in financial worries may be an improvement in the employment rate. Compared with a peak civilian unemployment rate of

14.7% in April of 2020 [36], the unemployment rate was 6% during March and April of 2021, steadily decreasing to 4% by January of 2022.

Additional policies introduced later on such as a tax credit to decrease the burden of paid leave on small businesses and non-profit organizations did not show any increase in correlation. It may be that by the end of the first wave, rather than nationwide policies, state-level government policies had more influence on the mental health of residents, and the direction each state took was not necessarily related to or dictated based on geographical regions or political party preference. We hypothesized that policies impacting a large number of people, such as eviction moratoriums and unemployment benefits ending would result in an increase in correlation values across all groups and states, however, the average correlations during both the second and third waves remain stable 30 days after these policies are implemented.

During the second wave period, we see an increase in the correlation between the Republican states in late July and a large spike in COVID-19 cases, hospitalization, and deaths in the states (see S2 Fig). This aligned with the period CDC announced the presence of the Delta variant. We observed a low correlation with no clear trend between the states regarding their political party preference during the third wave. However, the high number of deaths in the Republican states during the second wave persisted as the number of deaths in the states fluctuated.

For the regional dynamic connectome for worried about finances in Fig 6(c), we observed an initial high correlation (all above 0.7 except for the northeast with 0.45) at day 0 in the first 30 daysd uring the first wave. Looking at the reported values in Fig 2(c), the worried about finances indicator decreases rapidly across the regions from March 3rd to April 1st of 2021 (during the first 30 days). The high correlation value in the connectome reflects how all the states showed a decrease in the worried about finances indicator. However, there was a subsequent decrease in the correlation values (except for the southern states with a medium correlation peak of 0.4 around the 110—135 day marks) is assumed to imply that the states exhibit their own trends, and this interstate-level difference resulted in little to no correlation. This uptick was also observed in the west. It could be possible that states most heavily impacted by COVID-19 took similar methods to address the Delta variant, resulting in a brief increase in correlation among the southern states.

During the second wave the highest correlation value was observed in the southern states at the 110-day mark on the connectome in Fig 6(c). However, during the third wave, there was no clear correction between the states. The same policy and economic factors influencing the dynamic connectome for the political party preference during this period also affect the regional connectome.

## Comparison with Fulk et. al [22] results

Fulk et. al [22] investigated the effect that several variables related to the COVID-19 pandemic had on anxiety and depression in the general population during the same time period as this study. They presented their findings in a bidirectional format and found that variables directly related to COVID-19 (i.e. COVID-19 incidence and death) had a significant positive effect on anxiety and depression in several states. In our study, we found that the correlation in anxiety and depression between states increased dramatically in the second time period. This coincided with the rise of the Delta variant of SARS-CoV-2, which had significantly increased transmissibility compared to previously dominant strains of SARS-CoV-2.

One advantage of using sliding window and clustering analyses over the bidirectional approach taken in [22] is that we are able to see greater variability through time and across the country. Figs 1, 2, 5, and 6 in [22] show that an increase in the incidence of COVID-19 led to

an increase in the percentage of individuals that reported feeling anxious/depressed in a large number of states. We can make similar conclusions based on the results in Fig 2(a) and 2(b) in conjunction with Fig 4(a) and 4(b). Further, we can see which states were behaving similarly (e.g. the south cluster) in their correlation over the entire period of the analysis and we do not have to rely solely on significant results to draw our conclusions. An additional insight that the results of [22] pertains to why certain states are commonly clustered together. For example, Florida and Texas were significant in several of the analyses of the effect of COVID-19 incidence on anxiety and depression, which may indicate that their populations were responding similarly to the growing pandemic and thus explains why they are grouped together with respect to each of the variables evaluated in our study. The variables related to social isolation in [22] were less informative to our study and seemed to vary significantly more compared to variables related to COVID-19. This may indicate that the trends seen in our indicators were impacted more by COVID-19 related issues than social isolation.

## Conclusion

Numerous studies have observed an increase in negative mental health conditions across the United States in parallel with the emergence of COVID-19 [8–10, 22]. An increase in COVID-19 incidence and deaths has been implicated in increased feelings of anxiety and depression across multiple states [22]. In this work, we used clustering in network analysis to study the similarities between states and how COVID-19 impacts mental health across the country using the "mental health connectome" obtained from the covariance matrix of mental health indicators, such as worried about finances, feeling anxious and feeling depressed. We superimpose our correlation outcomes onto COVID-19 related data and implemented policies and mandates. Our results agree with numerous studies that observed an increase in negative mental health conditions across the United States in parallel with the emergence of COVID-19. We summarize our findings below:

(i) The southern states clustered together in the same group for feeling anxious, and worried about finances indicators. There were no identifiable clusters for feeling depressed indicator.

(ii) Increase in feeling anxious and feeling depressed in the southern states overlapped with the increase in COVID-19 related cases, deaths, and hospitalizations when the Delta variant was spreading rapidly.

(iii) The highest correlation values for the feeling anxious and feeling depressed variable seemingly overlapped with an increase in COVID-19 related cases, deaths, hospitalizations, and rapid spread of the COVID-19 delta variant.

(iv) Economic-driven policies and government intervention limit the burden of financial worries during the first wave but this was not the case during the second and third waves.

## Future steps and limitations

Results from this study showed a decrease in worries about finances variable across the country in the correlation among states and political affiliation; conducting further research using more survey data, particularly from dates earlier than March 3rd, 2021 may provide further insight into possible reasons for the decrease.

Anxiety can be fueled by sources such as media exposure and negative experiences with COVID-19; however, the government and state response to increasing cases may also alleviate or exacerbate feelings of anxiety. Since the data collected for this time frame began in the early

half of 2021, most lockdown measures had ended, and organizations such as the CDC were consistently issuing and updating health mandates and guidance information. In addition to national government responses, some states were actively loosening restrictions compared to others that maintained some level of safety measures to combat the appearance of COVID-19 variants. Whether participants felt safe or worried in response to loosening restrictions in that state could influence how they responded to the survey, affecting how a state related to other regions, and are a potential area for future research. Previous studies have identified certain protective factors against symptoms of anxiety, including old age, higher education, being male, and a good economic situation [37]. Comparing these factors with the map distribution for anxiety, such as the average level of education attained in each state, could provide more insight as well.

Given how closely certain policies were implemented, such as the expiration of an eviction moratorium as well as unemployment benefits happening within 11 days of each other, it is difficult to speculate how a given policy may have impacted a mental health indicator. It is also fully possible that the impact a policy could have on a variable—if any—arose due to the anticipation of a policy being introduced or a mandate being lifted rather than after it is implemented. Since we observed any correlation fluctuation 30 days after a policy was introduced rather than before, we have not taken anticipatory effects into consideration. In addition, by using a 30-day window, our sliding window analysis may have become less sensitive to short-term changes in correlation, as the values would need to collectively increase or decrease for a longer span of time in order to be picked up by the analysis.

Sliding window analysis requires the researchers to balance parameters such as window size prior to conducting the analysis. However, as there is no established standard for setting a window size, it is up to each researcher to choose a parameter that best fits their research question and intentions resulting in ambiguity. Incorporating different methods of centrality and network analysis would allow for more insight, stability, and interpretation of the changes in values over time, which the sliding window analysis and eigenvector centrality values may not fully provide.

Another limitation observed in this study is the difficulty of interpreting the feelings of depression variable. This mental health indicator is difficult to measure due to the long-term nature of depression, as well as the overlapping effects certain policy changes may have on the mental health indicators of interest. Unlike anxiety and worries about finances, depressive symptoms may take much longer to manifest themselves in any person, making it difficult to determine what policies impacted this indicator. Any singular policy change may also affect a multitude of mental health indicators, and the extent to which each indicator is affected cannot readily be analyzed.

## Supporting information

**S1 Fig. Mental health indicators based on political party preference.** Percentage of individuals (a) feeling anxious, (b) feeling depressed, and (c) worried about finances.
(TIF)

**S2 Fig. COVID-19 related data based on political party preference.** Daily COVID-19 (a) cases, (b) reported deaths, and (c) hospitalization cases.
(TIF)

## Author Contributions

**Conceptualization:** Folashade B. Agusto.

**Formal analysis:** Hiroko Kobayashi, Alexander Fulk.

**Funding acquisition:** Folashade B. Agusto.

**Supervision:** Folashade B. Agusto.

**Visualization:** Hiroko Kobayashi, Alexander Fulk, Folashade B. Agusto.

**Writing – original draft:** Hiroko Kobayashi, Raul Saenz-Escarcega, Alexander Fulk.

**Writing – review & editing:** Hiroko Kobayashi, Raul Saenz-Escarcega, Alexander Fulk, Folashade B. Agusto.

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
