## [Decision Letter · Decision Letter 0]

23 Feb 2023

PONE-D-22-28525Understanding mental health trends during COVID-19 pandemic in the United States using network analysisPLOS ONE

Dear Dr. Augusto

Thank you for submitting your manuscript to PLOS ONE. After careful consideration, we feel that it has merit but does not fully meet PLOS ONE’s publication criteria as it currently stands. Therefore, we invite you to submit a revised version of the manuscript that addresses the points raised during the review process.

We look forward to receiving your revised manuscript.

Kind regards,

Karina Cardoso Meira, Ph.D

Academic Editor

PLOS ONE

Journal Requirements:

This research was funded by the National Science Foundation, grant number DMS2028297 and DMS2230117.

However, funding information should not appear in the Acknowledgments section or other areas of your manuscript. We will only publish funding information present in the Funding Statement section of the online submission form. 

This research was funded by the National Science Foundation, grant number DMS2028297 and DMS2230117.

4. We note that Figures 2, 4 and 5 in your submission contain [map/satellite] images which may be copyrighted. All PLOS content is published under the Creative Commons Attribution License (CC BY 4.0), which means that the manuscript, images, and Supporting Information files will be freely available online, and any third party is permitted to access, download, copy, distribute, and use these materials in any way, even commercially, with proper attribution. For these reasons, we cannot publish previously copyrighted maps or satellite images created using proprietary data, such as Google software (Google Maps, Street View, and Earth). For more information, see our copyright guidelines: http://journals.plos.org/plosone/s/licenses-and-copyright.

a. You may seek permission from the original copyright holder of Figures 2, 4 and 5 to publish the content specifically under the CC BY 4.0 license.  

Additional Editor Comments:

This manuscript has an essential theme. I liked reading it. Make changes to the description of the methodology and results so that these sections are easy to understand for readers who need to gain expertise in the methods used.

Reviewers' comments:

Reviewer's Responses to Questions

**Comments to the Author**

1. Is the manuscript technically sound, and do the data support the conclusions?

Reviewer #1: Yes

Reviewer #2: Partly

2. Has the statistical analysis been performed appropriately and rigorously? 

Reviewer #1: I Don't Know

Reviewer #2: Yes

3. Have the authors made all data underlying the findings in their manuscript fully available?

Reviewer #1: Yes

Reviewer #2: Yes

4. Is the manuscript presented in an intelligible fashion and written in standard English?

Reviewer #1: Yes

Reviewer #2: Yes

5. Review Comments to the Author

Reviewer #1: Dear authors,

My congratulations for the quality of the manuscript and political and psychosocial approach to the impact of COVID-19 on the American population. It is certainly a type of study to be replicated in other countries.

Reviewer #2: This is a very relevant paper with a very important thematic. I liked reading it. The text flows very nicely, and the relevance of the study is emphasized throughout the paper. One thing that makes me feel uncomfortable overall is that I am not a specialist on the methods. The interpretations are hard, and the methodological discussion too. For the readers of Plos One, an interdisciplinary journal, it may be an obstacle for the reader. The interpretation of the results is compromised because of its difficulty, as it follows from the methodology section. Nevertheless, I leave it to the editor for a consideration, giving the preference of current readers.

Bearing my limitations in mind, I have a few considerations:

define "connectome" in the abstract

define "connectome" for those not familiar with the area of study in the very fisrt appearance on the paper.

dessazonalize all data for the analysis - sazonality is a challenge for quantitative methods. It usually follows as a standard procedure to dessazonalize data in order to have the meaningful curvatures in the analysis.

list on page 7 could be translated into a box, according to "Policy guideline that affects finances" and "Policy guideline that affects feeling anxious and depressed"

Figure 4: Country-wide maps grouping states with similar trends between 3/2/2021 to 1/10/2022 for (a) feeling anxious, (b) feeling depressed and (c) worried about finances indicators. The states with light blue, blue, and dark blue are in the same cluster.  so that means that all country is in the same cluster?

I could find out on the subtitle for figure 5 that my comment below its not the case. So please change figure's 4 subtitle.

6. PLOS authors have the option to publish the peer review history of their article (what does this mean?). If published, this will include your full peer review and any attached files.

Reviewer #1: No

Reviewer #2: **Yes: **Raquel Guimaraes

---

## [Author Response · Author response to Decision Letter 0]

26 Mar 2023

The responses have been included as an attached file.

---

## [Decision Letter · Decision Letter 1]

25 May 2023

Understanding mental health trends during COVID-19 pandemic in the United States using network analysis

PONE-D-22-28525R1

Dear Dr. Agusto

We’re pleased to inform you that your manuscript has been judged scientifically suitable for publication and will be formally accepted for publication once it meets all outstanding technical requirements.

Kind regards,

Karina Cardoso Meira, Ph.D

Academic Editor

PLOS ONE

Additional Editor Comments (optional):

Dear Dr. Agusto

We’re pleased to inform you that your manuscript has been judged scientifically suitable for publication and will be formally accepted for publication once it meets all outstanding technical requirements.

Reviewers' comments:

Reviewer's Responses to Questions

**Comments to the Author**

1. If the authors have adequately addressed your comments raised in a previous round of review and you feel that this manuscript is now acceptable for publication, you may indicate that here to bypass the “Comments to the Author” section, enter your conflict of interest statement in the “Confidential to Editor” section, and submit your "Accept" recommendation.

Reviewer #2: All comments have been addressed

2. Is the manuscript technically sound, and do the data support the conclusions?

Reviewer #2: Yes

3. Has the statistical analysis been performed appropriately and rigorously? 

Reviewer #2: Yes

4. Have the authors made all data underlying the findings in their manuscript fully available?

Reviewer #2: Yes

5. Is the manuscript presented in an intelligible fashion and written in standard English?

Reviewer #2: Yes

6. Review Comments to the Author

Reviewer #2: The authors addressed the reviewers comments in a very timely and comprehensive manner. I believe the article is now ready for publishing.

7. PLOS authors have the option to publish the peer review history of their article (what does this mean?). If published, this will include your full peer review and any attached files.

Reviewer #2: **Yes: **Raquel Guimaraes

---

## [Editor Report · Acceptance letter]

30 May 2023

PONE-D-22-28525R1 

Understanding mental health trends during COVID-19 pandemic in the United States using network analysis 

Dear Dr. Agusto:

I'm pleased to inform you that your manuscript has been deemed suitable for publication in PLOS ONE. Congratulations! Your manuscript is now with our production department. 

Kind regards, 

on behalf of

Dr. Karina Cardoso Meira 

Academic Editor

PLOS ONE